# Dynamic Analysis and Experiment of Multiple Variable Sweep Wings on a Tandem-Wing MAV

Liang Gao [1], Yanhe Zhu [1,*], Xizhe Zang [1,*], Junming Zhang [1], Boyang Chen [1], Liyi Li [2] and Jie Zhao [1]

1 State Key Laboratory of Robotics and System, Harbin Institute of Technology, Harbin 150001, China; gaoliang@hit.edu.cn (L.G.); 19b908071@stu.hit.edu.cn (J.Z.); 19b908065@stu.hit.edu.cn (B.C.); jzhao@hit.edu.cn (J.Z.)
2 School of Electrical Engineering and Automation, Harbin Institute of Technology, Harbin 150001, China; liliyi@hit.edu.cn
* Correspondence: yhzhu@hit.edu.cn (Y.Z.); zangxizhe@hit.edu.cn (X.Z.)

**Abstract:** The current morphing technologies are mostly regarded as auxiliary tools, providing additional control torques to enhance the flight maneuverability of unmanned aerial vehicles (UAVs), and they cannot exist independently of the traditional control surfaces. In this paper, we propose a tandem-wing micro aerial vehicle (MAV) with multiple variable-sweep wings, which can reduce the additional inertia forces and moments and weaken the dynamic coupling between longitudinal and lateral motion while the MAV morphs symmetrically for pitch control or asymmetrically for roll control, thereby flying without the traditional aileron and elevator. First, load experiments were conducted on the MAV to verify the structural strength of the multiple variable sweep wings, and the control moments caused by the morphing of the MAV were presented through numerical simulations. Then, the effects caused by symmetric and asymmetric morphing were investigated via dynamic response simulations based on the Kane dynamic model of the MAV, and the generated additional inertia forces and moments were also analyzed during morphing. Finally, dynamic response experiments and open-loop flight experiments were conducted. The experimental results demonstrated that the morphing mode in this study could weaken the coupling between the longitudinal and lateral dynamics and that it was feasible for attitude control without the traditional aileron and elevator while flying.

**Keywords:** tandem-wing; variable sweep; morphing aircraft; dynamic modeling; additional inertia force



## 1. Introduction

The rapid development of drones not only provides an ideal experimental platform for various emerging technologies but also allows researchers to design new UAVs with stronger maneuverability and higher efficiency without considering the physiological limits of pilots [1]. Morphing technology enables drones to adaptively change the shape of their wings or fuselage according to different flight tasks or environments, just like birds in nature, in order to obtain the most suitable flight performance [2,3]. Morphing technology can improve the flight efficiency of aircraft, expand the flight envelope, increase the use of aircraft, or provide aircraft attitude control [4–6]. Therefore, research on the designing of UAVs with morphing technology has been gradually increasing [7].

Variable sweep is a commonly used morphing technique with a significant amount of research dedicated to it; examples include multiple joint variable sweep aircraft morphing (like birds) [8] and symmetrical sweep morphing for pitch control [9]. The AquaMAV can dive from the air to collect water samples underwater and then drill out of the surface to take off again. The skill is achieved by wings sweeping backward to reduce water resistance during the processes diving and taking off from the water [10]. Wings sweeping forward can generate nose-up moment, which is used to enhance perching maneuvers [11]. Variable

sweep is also explored to realize roll control via asymmetric morphing, and it has been proven to have better performance at high angles of attack [12]. In addition, there are also many other types of morphing UAVs. A transformer aircraft with span morphing can morph symmetrically to enhance flight performance and asymmetrically to provide roll control [13]. Vos et al. designed a new type of active twisting wing that can enhance the lift–drag ratio in various flight conditions [14]. Barrett et al. proposed a new type of adaptive structure relying on pressurized honeycomb cells; by varying the cell pressure, the flap changed its geometry and subsequently altered the lift coefficient [15]. Bishay et al. proposed a parametric study of a composite skin for a twist-morphing wing [16]. Obradovic et al. presented a simulation methodology for a morphing gull-wing aircraft [17]. Han et al. used smart soft composite to fabricate morphing winglets, and when the morphing winglet was actuated, the lift–drag ratio increased compared with the flat wing geometry [18]. The endless morphing technologies, such as variable camber [19], variable thickness [20], active wing [21], and so on, all utilize wing morphing to achieve drone maneuverability control or enhance the flight performance. Another bionic morphing wing drone, which resembles a bird in appearance but flies without flapping wings, uses discrete feather-like plates for morphing surfaces and extends drone flight capabilities through wing and tail morphing [22].

From the literature above, it is observed that morphing technology changes the shape of the UAV's wings or fuselage through morphing mechanisms or material. Symmetric morphing only causes changes in the longitudinal movement of the UAV, achieving pitch control or multitask requirements; whereas asymmetric morphing is usually used for roll control. Nevertheless, the studies on asymmetric morphing are relatively few; this is probably due to the coupling of lateral and longitudinal dynamics caused by asymmetric morphing, which will increase the burden on the control system [23]. In addition, morphing not only brings significant changes in aerodynamic forces and gravity moments [24] but also produces additional inertia forces and moments, especially at the onset and end of a morphing maneuver, which can even cause instability during morphing [17,25]. However, UAV research has focused on resolving aerodynamic characteristics, while inertia properties such as inertia force and inertia moment during morphing are seldom addressed [26].

Currently, most studies on morphing UAVs ignore the dynamic effects of morphing [5,6,12,17,25] and are working on enhancing the robustness of the control system in restraining dynamic disturbances [24,25]. In addition, morphing, as an auxiliary tool, provides additional control torques to enhance the flight maneuverability of the UAV, while the flight stability of the UAV is realized by the traditional control surfaces under the action of the control system [6]. In other words, the current morphing technology cannot exist independently of the traditional control surfaces. The bionic morphing wing drone is the only one without traditional control surfaces, but the morphing tail resembles a fully active tail which can not only morph but also be used as an elevator and rudder, making their structure more complex and weaker in strength [22]. In addition, there are only a few morphing methods that can simultaneously achieve roll and pitch attitude control, but due to the coupling between the longitudinal and lateral dynamics when the aircraft morphs asymmetrically, the elevator and aileron are also indispensable [6,12].

Inspired by variable sweep technology, a tandem-wing MAV with multiple variable-sweep wings is proposed. With backward-swept canards and forward-swept wings, the MAV can achieve longitudinal stabilization. Moreover, due to the opposite sweeping directions of the canards and the wings, the additional inertia forces and moments can be counteracted during morphing, reducing the coupling between the lateral and longitudinal dynamic. This design allows for the complete replacement of the elevator and aileron and is particularly suitable for the popular tube-launched tandem-wing UAV design [27,28], which has better aerodynamic performance at high speeds and can accomplish multitask adaptability [29]. In this paper, we studied the dynamic characteristics of the morphing process and developed a prototype for flight testing. The main contributions of this paper are the following:

(1) The multiple variable sweep design can reduce the additional inertia forces and moments and weaken the dynamic coupling between longitudinal and lateral motion during morphing;

(2) The tandem-wing MAV can morph symmetrically for pitch control and asymmetrically for roll control and fly without traditional aileron and elevator.

The sections of the paper are organized as followings. In Section 2, the design and load test of the tandem-wing MAV with multiple variable sweep wings are presented, and the aerodynamic characteristics with morphing are investigated. In Section 3, the longitudinal and lateral dynamic models are decoupled and simplified, and the effects on the MAV caused by symmetric and asymmetric morphing are investigated through dynamic response simulation. In Section 4, dynamic response experiments and open-loop flight experiments are conducted. Finally, conclusions based on the presented results are presented in Section 5.

## 2. Design of the Tandem-Wing MAV and Aerodynamic Characteristics

### 2.1. Design of the Tandem-Wing MAV

The schematic diagram of the tandem-wing MAV with multiple variable sweep wings is presented in Figure 1. $O_b x_b y_b z_b$ is the body coordinate system, and $O_b$ is located at the center of mass of the fuselage. The $x_b$-axis points in the forward direction of the MAV nose, the $y_b$-axis points to the right side of the MAV, and the $z_b$-axis is determined by the right-hand rule. The MAV is composed of a propulsion system, a fuselage, a pair of canards and wings, four sets of morphing driving mechanisms, a folding vertical tail, an electronic bay, and other components, and there is no conventional elevator or aileron. The four sets of morphing driving mechanisms are four-bar mechanisms which drive the canards and the wings by servos with connecting rods, respectively. The MAV can be controlled by changing the lift distribution through the symmetrical or asymmetrical sweep morphing of the canards and wings according to mission instructions during flight. The canards can only sweep backward and the wings can only sweep forward, and considering the aerodynamic performance, the range of sweep angle is limited to from 0° to 30°. The main parameters of the tandem-wing MAV are shown in Table 1.

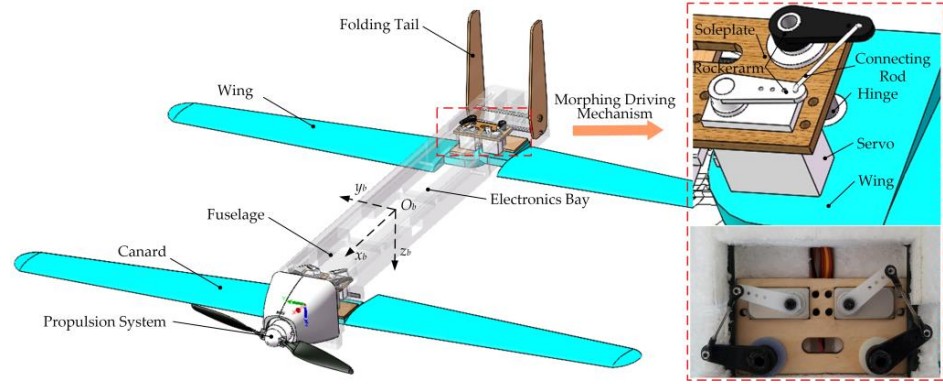

**Figure 1.** Schematic diagram of the tandem-wing MAV with multiple variable sweep wings.

**Table 1.** Parameters of the tandem-wing MAV.

| Parameter | Unit | Value |
|---|---|---|
| Length of fuselage, $l_f$ | m | 0.72 |
| Span, $b$ | m | 0.893 |
| Mass of the MAV, $m$ | kg | 1.67 |
| Stagger between canard and wing, $St$ | m | 0.42 |
| Mean aerodynamic chord, $c_A$ | m | 0.077 |
| Reference area, $S$ | m$^2$ | 0.135 |
| Design flight velocity, $V$ | m·s$^{-1}$ | 20–30 |

A prototype of the tandem-wing MAV was manufactured using lightweight expanded polypropylene (EPP) material embedded with carbon fiber tubes for reinforcement, as shown in Figure 2a. The fuselage must have sufficient strength to support the wing flight loads while ensuring the wings can perform sweep morphing with enough stability. A wooden board was placed at the junction of the wing and the fuselage to enlarge the contact area while providing enough support force to wing.

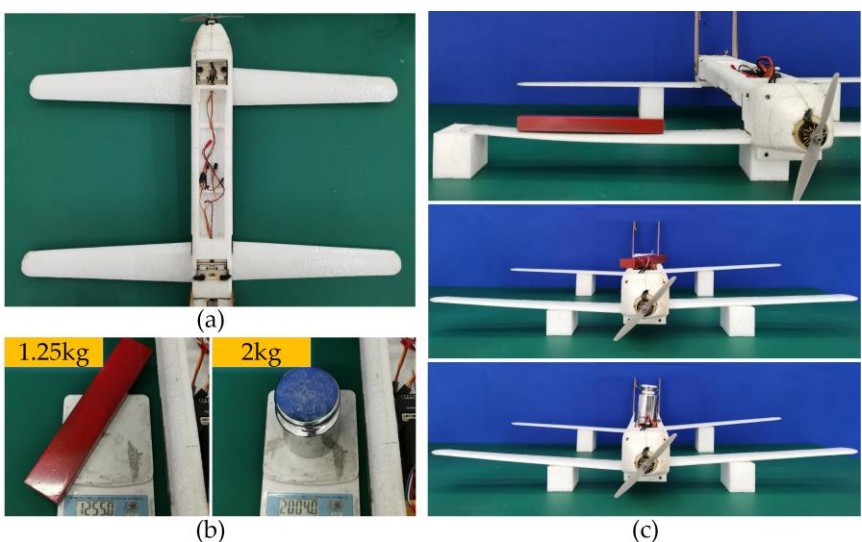

**Figure 2.** Prototype and load test of the tandem-wing MAVL: (**a**) prototype of the tandem-wing MAV; (**b**) clump weight and weights; (**c**) load test of the MAV.

The wing loading was inspected via body load experiments. The mass of clump weight and weights were 1.25 kg and 2 kg, respectively, as shown in Figure 2b. During full-load flight, a single wing only bears about one-quarter of the body weight. From the results shown in Figure 2c, it can be seen that the deformations of the wing were not significant, even after adding clump weight that was much greater than the load that a single wing typically bore during flight. This indicated that the wings had adequate wing loading. When a 1.25 kg clump weight was attached to the fuselage in the load experiment, the deformations of the wings were not significant, as shown in Figure 2c. When a 2 kg weight was added, which was more than 1.2 times the flight overload, the wings only underwent slight deformation. Therefore, the wing loading can meet the strength requirements of the MAV. The wing support points shown in the figures were located at the aerodynamic centers of each wing.

### 2.2. Aerodynamic Characteristics with Morphing

Symmetric sweep morphing refers to the simultaneously sweeping backward of the two canards or forward of the two wings, which is used for pitch control of the MAV. Asymmetric sweep morphing refers to the simultaneously sweeping backward of the canard and forward of the wing on the same side, which is used for roll control. $h_i$ is the sweep angle of the airfoils; and $i = 1, 2, 3,$ and 4 denote the left canard, right canard, left wing, and right wing, respectively; $\delta_e$ and $\delta_a$ are defined as control inputs for symmetric morphing and asymmetric morphing, respectively; and $\delta_e$ replaces the elevator function, while $\delta_a$ replaces the aileron function. This mode is similar to the elevator and aileron configuration of a flying wing without the presence of a yaw channel, and the turning is achieved through the method of bank-to-turn. $\delta_e$ and $\delta_a$ are expressed using the following formula, and $\delta_e, \delta_a \in [-30°, 30°]$.

$$\delta_e = \begin{cases} h_1 = -h_2, \ h_3 = h_4 = 0, \ \delta_e < 0 \\ h_3 = -h_4, \ h_1 = h_2 = 0, \ \delta_e > 0 \end{cases} \tag{1}$$

$$\delta_a = \begin{cases} h_1 = -h_3, \ h_2 = h_4 = 0, \ \delta_a < 0 \\ h_2 = -h_4, \ h_1 = h_3 = 0, \ \delta_a > 0 \end{cases} \quad (2)$$

The aerodynamic characteristics of the tandem-wing MAV while morphing have been acquired via a computational fluid dynamics (CFD) approach [30], and the pitch moment coefficient and roll moment coefficient at different morphing configurations are shown in Figure 3. All the aerodynamic parameters are calculated based on the body coordinate system $O_b x_b y_b z_b$. Figure 3a shows the variation of pitch moment coefficient $C_m$ of the MAV with attack angle $\alpha$ when the sideslip angle $\beta = 0°$. In this case, the lateral force, rolling moment, and yawing moment are all zero. From the graph, it can be observed that symmetric morphing can significantly generate pitch moment. In addition, the control moment increases with the increase in $\alpha$. As $\delta_e$ increases, the slope of the pitch moment coefficient also increases, indicating that a larger morphing leads to a stronger control moment. When $\alpha = 4°$, the maximum pitch-up and pitch-down moments generated by symmetric morphing are 0.618 N·m and $-1.21$ N·m, respectively. Figure 3b shows the variation of roll moment coefficient $C_l$ of the MAV with sideslip angle $\beta$ when $\alpha = 4°$. Due to the symmetry, we only consider the situation of the left side morphing; thus, the range of $\delta_a$ is selected from $-30°$ to $0°$ in the figure. It can be observed that morphing of the left side can significantly generate leftward roll moments, which is because the lift on the left side decreases after morphing. As $\delta_a$ increases, the roll moment coefficient also increases, indicating that a larger morphing leads to a stronger control moment. When $\beta = 0°$, the maximum roll moment generated by asymmetric morphing can reach $-0.487$ N·m. Here, the minus sign indicates the direction of the moments and angles in a body coordinate system based on right-hand rule. As $\beta$ increases, the roll moment gradually increases, but the change is not significant.

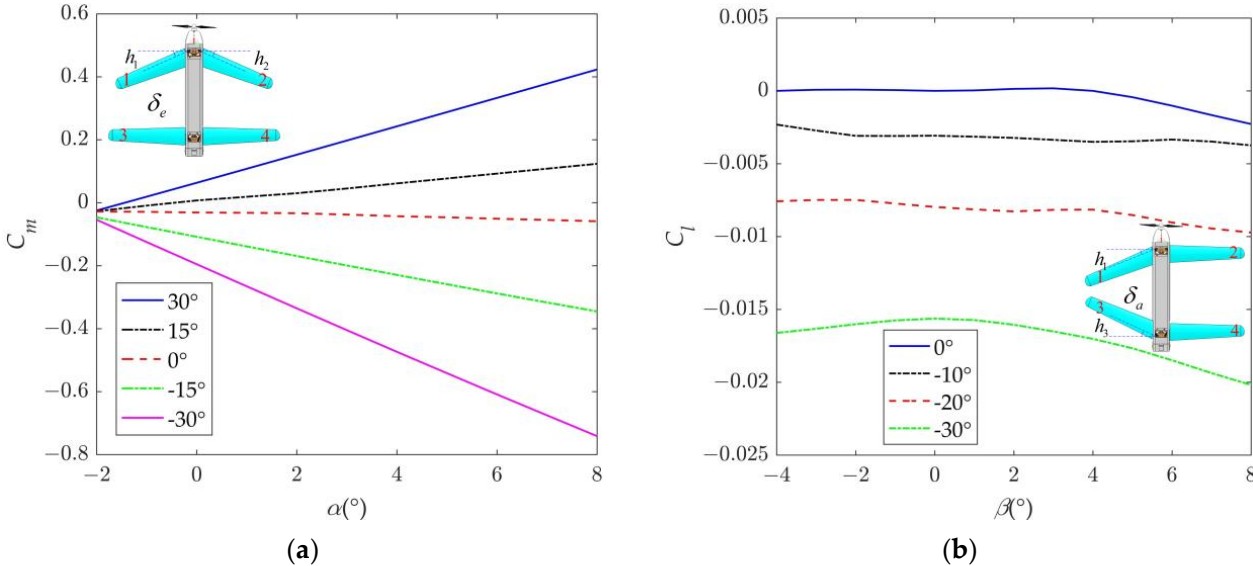

**Figure 3.** Control moment coefficients of the tandem-wing MAV while morphing: (**a**) pitch moment coefficient change with $\alpha$ at different $\delta_e$ ($\beta = 0°$); (**b**) roll moment coefficient change with $\beta$ at different $\delta_a$ ($\alpha = 4°$).

## 3. Modeling and Dynamic Analysis

The conventional aircraft can be regarded as a six-degrees-of-freedom system, while the tandem-wing MAV in our paper has four sweep wings, each of which can rotate in the plane, making it a more complex ten-degrees-of-freedom multi-rigid-body system. The large scale and rapid morphing of the MAV will cause changes in parameters such as inertia forces, aerodynamic forces, moments of inertia, center of mass, and so on. In order to accurately describe the dynamic model of the tandem-wing MAV and study the dynamic

characteristics affected by morphing, the nonlinear dynamic model of the tandem-wing MAV has been built via the Kane method and is shown in our previous research [30]. We will decouple and simplify the longitudinal and lateral dynamic models from the Kane dynamic equations and investigate the effects on the MAV caused by symmetric and asymmetric morphing.

### 3.1. Longitudinal Dynamic Analysis with Symmetric Morphing

3.1.1. Model Simplification of Longitudinal Motion

When the MAV undergoes symmetric morphing, it will cause a change in pitch moment and result in longitudinal motion response. Considering the level flight without sideslip of the MAV, where the sideslip angle $\beta$ and roll angle $\phi$ satisfy $\beta = \phi = 0$, plugging Equation (1) into the Kane dynamic equations of the MAV and simplifying, the dynamic model of the MAV under symmetric morphing can be obtained as follows:

$$\begin{cases} F_x + F_{x\delta_e} = m(\dot{u} + wq) \\ F_z + F_{z\delta_e} = m(\dot{w} - uq) - 2m_a\dot{q}(a_c - a_w) \\ M + M_{\delta_e} + M_G = J_y\dot{q} - 2m_a(\dot{w} - uq - g\cos\theta)(a_c - a_w) \end{cases}, \tag{3}$$

where $m$ is the mass of the entire MAV and $m_a$ is the mass of a single airfoil; $u$ and $w$ are the flight velocity in the $x_b$- and $z_b$-axis of the body coordinate frame, respectively; $\theta$ is the pitch angle; $q$ is the pitch rate; and $M$ is the control pitch moment.

$F_x$, $F_y$, and $F_z$ are forces in the $x_b$-, $y_b$-,and $z_b$-axis of the body coordinate frame, respectively, and they can be expressed as follows:

$$\begin{cases} F_x = P - mg\sin\theta - D\cos\alpha\cos\beta - Y\cos\alpha\sin\beta + L\sin\alpha \\ F_y = mg\cos\theta\sin\phi - D\sin\beta + Y\cos\beta \\ F_z = mg\cos\theta\cos\phi - D\sin\alpha\cos\beta - Y\sin\alpha\sin\beta - L\cos\alpha \end{cases}, \tag{4}$$

where $P$, $L$, $D$, and $Y$ are the thrust, lift, drag, and side force of the MAV, respectively. Here, considering the situation of level flight without sideslip and symmetric morphing, $F_y$ is equal to zero.

$F_{x\delta_e}$, $F_{z\delta_e}$, $M_{\delta_e}$, and $M_G$ are the additional inertia force in the $x_b$-axis, the additional inertia force in the $z_b$-axis, the additional inertia pitch moment caused by morphing, and the gravity moment generated by the mass center shift, respectively. The detailed expressions are represented in Appendix A, Equations (A1) and (A2).

Symmetric morphing will only cause a gravity center change in the $x_b$-axis; the shift of gravity center $\Delta x_{cg}$ is expressed as

$$\Delta x_{cg} = \frac{2m_a l \sin\delta_e}{m}, \tag{5}$$

where $l$ is the distance between airfoil mass center and rotation center, $l = 0.14$ m. Because $\delta_e \in [-30°, 30°]$, $\Delta x_{cg}$ does not exceed $\pm 7$ mm.

Symmetric morphing does not generate lateral aerodynamic forces and moments and does not cause changes in lateral state variables. Therefore, the lateral state variables in the nonlinear dynamic model are all equal to zero, and Equation (3) is the longitudinal dynamic equation of the tandem-wing MAV. Compared with the traditional dynamic model of aircraft, the symmetric morphing of the MAV not only leads to changes in the wing position but also to changes in the gravity center, resulting in a tiny pitch moment change opposite to the required control moment. At the same time, the rotation speed and acceleration of the wings will also produce additional inertia forces and moments, which disappear when the wings stop rotating. The existence of additional inertia forces and moments makes the dynamics of the MAV complex and even affects its stability. Further analysis is required in order to understand the effects of these additional inertia forces and moments.

### 3.1.2. Dynamic Response Analysis of Symmetric Morphing

Based on the established longitudinal dynamics model Equation (3), the state variables changes of the MAV during symmetric morphing were examined, and the additional inertia forces and moment effects caused by symmetric morphing were analyzed to determine their impact on the motion of the MAV. This analysis has practical significance for the control design of the MAV. Assuming that the MAV is in steady level flight with an initial state of $V = 20$ m/s and $\alpha = 4°$, the following cases were analyzed mainly to examine the influence of different morphing rates and to compare the results with those without additional inertia forces and moments effects.

case I: $\omega_n = 20.82$, $\zeta_n = 0.7$;
case II: $\omega_n = 41.63$, $\zeta_n = 0.7$;
case III: $\omega_n = 83.26$, $\zeta_n = 0.7$;
case IV: $\omega_n = 41.63$, $\zeta_n = 0.7$, without additional inertia forces and moments effects,

where $\omega_n$ and $\zeta_n$, respectively, are the natural frequency and the damping ratio of the wing rotation actuator, which is approximate to a second-order system [30] in which the corresponding actuator rise time of case I, II, and III are 0.2 s, 0.1 s, and 0.05 s, respectively. The changes in the motion parameters of the MAV under the symmetric morphing response are shown in Figure 4.

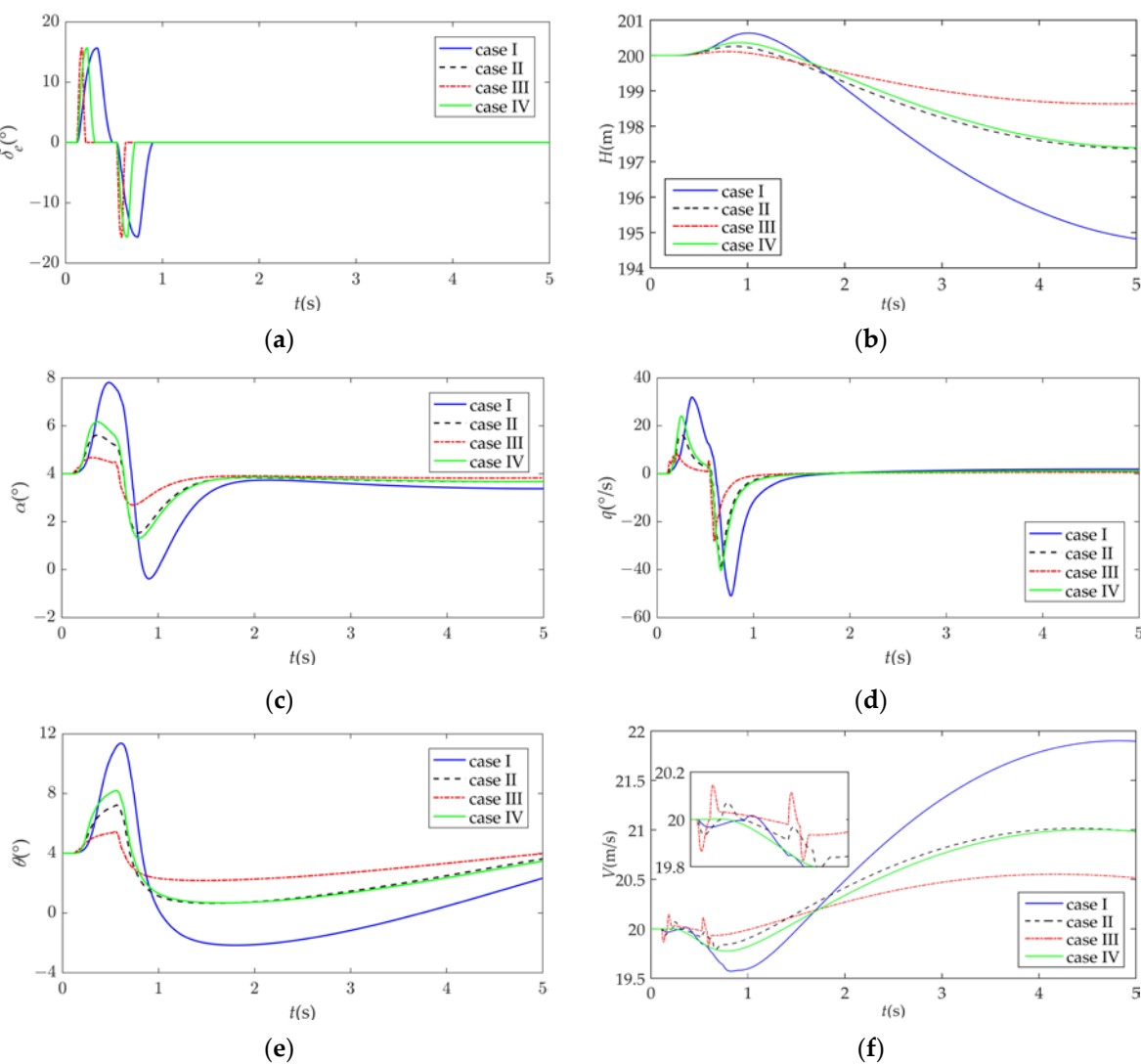

**Figure 4.** Dynamic responses of motion parameters caused by symmetric morphing: (**a**) $\delta_e$; (**b**) $H$; (**c**) $\alpha$; (**d**) $q$; (**e**) $\theta$; (**f**) $V$.

Figure 4a shows the morphing variation of the MAV. At the initial flight state, the wings begin to sweep forward 15° at 0.1 s and then recover; later, the canards begin to sweep backward 15° at 0.5 s and then recover. It can be seen that the various motion parameters of the MAV can quickly follow the morphing response. When $\delta_e > 0$, that is, the wings sweep forward, although the gravity center of the MAV moves forward, the positive pitch moment generated by morphing causes $\alpha$ and $\theta$ to increase, and the MAV begins to climb. On the contrary, when $\delta_e < 0$, that is, the canards sweep backward, pitch rate $q$ quickly turns negative, $\alpha$, $\theta$, and flight altitude $H$ all begin to decrease, and then the MAV begins to pitch down. After the morphing ends, flight speeds $V$ and $H$ enter a phugoid oscillation state. In addition, the faster the morphing rate, the faster the response speed of the various motion parameters of the MAV, but the smaller the change amplitude, because the faster the morphing rate, the shorter the action time of the dynamic aerodynamic force caused by morphing.

During MAV morphing, $V$ and $q$ have obvious fluctuation changes, especially $V$, and the faster the morphing rate, the more obvious the performance, as shown in Figure 4f. After the morphing ends, the changes gradually ease. However, the response results of Case IV do not show this performance, indicating that the fluctuation is the result of the additional inertia forces and inertia moments caused by morphing. In addition, the change in the amplitude of each motion parameter in Case II is slightly smaller than the response results of Case IV, indicating that the additional inertia forces and inertia moments have a certain inhibitory effect on the response of the motion parameters of the MAV.

In view of the influence of the additional inertia forces and moments, the change laws of the additional inertia forces and moments caused by symmetric morphing with different morphing rates were studied, and the results are shown in Figure 5. It can be seen that the inertia forces and moments caused by symmetric morphing increase with the increase in morphing rate. Among them, a larger additional inertia force $F_{x\delta_e}$ is generated in the $x_b$-axis, with a peak value exceeding 40 N, which is much larger than the aerodynamic force borne by the MAV itself, and according to Equation (A1), this is mainly due to the incipient acceleration of the actuator. Then, due to the existence of the distance between the mass center of airfoil and of fuselage in the $z_b$-axis, the additional inertia pitch moment $M_{\delta_e}$ is more than 0.6 N·m, which has the same order of magnitude as the pitch control moment generated by morphing. The additional inertia force $F_{z\delta_e}$ in the $z_b$-axis is very small, with a range of no more than ±0.2 N; this is because the wing rotation plane is perpendicular to the $z_b$-axis. The change in the gravity moment $M_G$ does not exceed ±0.06 N·m because a single airfoil accounts for a small proportion of the mass of the whole MAV, and the range of the gravity center shift caused by morphing is not large. The change in the gravity moment is not related to the morphing rate, only to the morphing amount. At the end of the morphing, the additional forces and moments both become zero.

In fact, at the onset and end of morphing, the inertia forces and inertia moments have obvious protrusions because they mainly depend on the acceleration of the actuator. However, due to their changes from a positive peak value to a negative peak value, the inertia forces and moment effects cancel each other out, so the overall effect on the motion of the MAV is relatively weak, but it will cause fluctuations in the state variables of the MAV. If the MAV needs to control the attitude by continuously morphing quickly, the influence of the inertia forces and inertia moments will be correspondingly amplified, which must be considered in the control design.

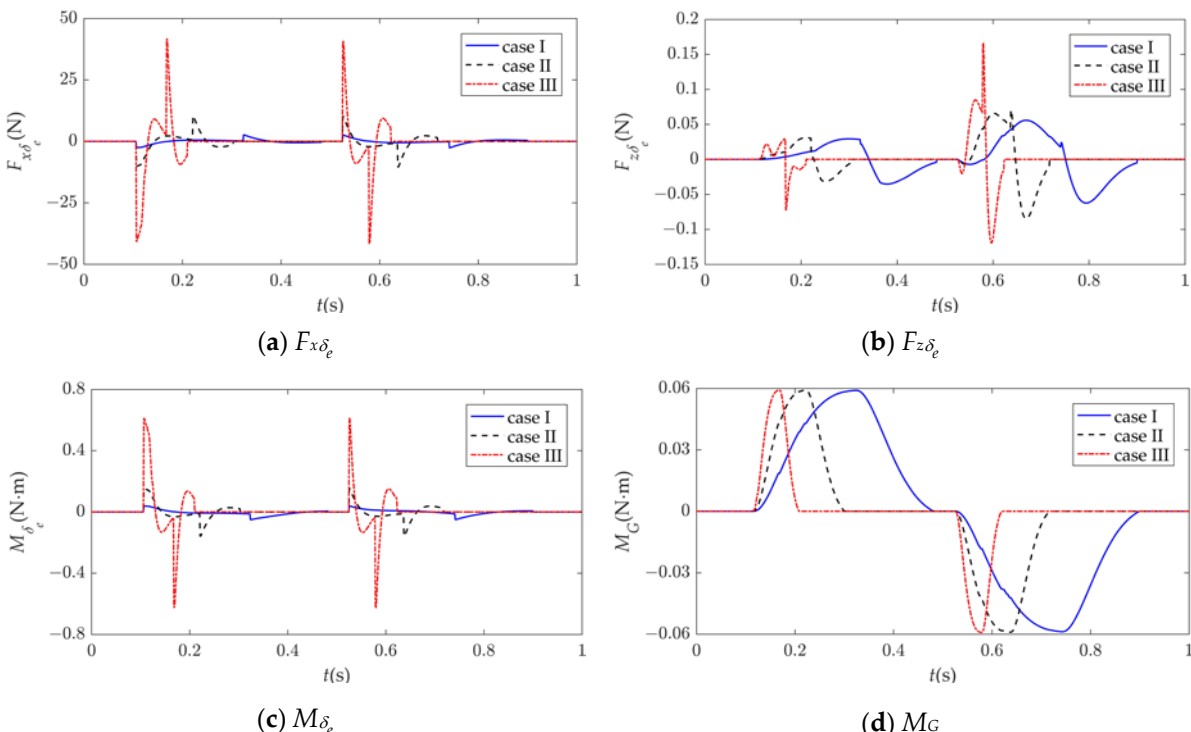

**Figure 5.** Additional inertia forces and moments caused by symmetric morphing: (**a**) $F_{x\delta_e}$; (**b**) $F_{z\delta_e}$; (**c**) $M_{\delta_e}$; (**d**) $M_G$.

*3.2. Lateral Dynamic Analysis with Asymmetric Morphing*

3.2.1. Model Simplification of Lateral Motion

This article proposes using the asymmetric morphing of the tandem-wing MAV to replace the traditional aileron control for the rolling attitude. The asymmetric morphing will cause the MAV to undergo coupled motion in all three axes. By using the planned asymmetric morphing derived from Equation (1) and inserting it into the Kane dynamic equations of the MAV, after simplifying, the dynamic model of the MAV under asymmetric morphing can be obtained as follows:

$$
\begin{cases}
F_x + F_{x\delta_a} = m(\dot{u} + wq - vr) \\
F_y + F_{y\delta_a} = m(\dot{v} + ur - wp) + 2m_a\dot{r}(a_c - a_w) \\
F_z + F_{z\delta_a} = m(\dot{w} + vp - uq) - 2m_a\dot{q}(a_c - a_w) \\
\overline{L} + \overline{L}_{\delta_a} + \overline{L}_G = J_x\dot{p} + J_1qr - [J_{xz}^f + 2m_ac(a_c + a_w)]\dot{r} - J_{xz}^f pq \\
M + M_{\delta_a} - 2m_ag\cos\theta\cos\varphi(a_c - a_w) = J_y\dot{q} + J_2pr + J_{xz}^f(p^2 - r^2) - 2m_a(\dot{w} + vp - uq)(a_c - a_w) \\
N + N_{\delta_a} + N_G + 2m_ag\cos\theta\sin\varphi(a_c - a_w) = J_z\dot{r} + J_3pq - J_{xz}^f(\dot{p} - qr) + 2m_a(\dot{v} + ur - wp)(a_c - a_w) - 2m_ac(a_c + a_w)\dot{p}
\end{cases} , \quad (6)
$$

where $v$ is the flight velocity in the $y_b$-axis of the body coordinate frame, respectively; $\phi$ is the roll angle; $p$ and $r$ are the roll rate and yaw rate, respectively; $\overline{L}$, $M$, and $N$ are the control roll moment, pitch moment, and yaw moment generated by morphing, respectively.

$F_{x\delta_a}$, $F_{y\delta_a}$, $F_{z\delta_a}$, $\overline{L}_{\delta_a}$, $M_{\delta_a}$, $N_{\delta_a}$, $\overline{L}_G$, and $N_G$ are the additional inertia force in the $x_b$-axis, the additional inertia force in the $y_b$-axis, the additional inertia force in the $z_b$-axis, the additional inertia roll moment, the additional inertia pitch moment, the additional inertia yaw moment caused by asymmetric morphing, the roll moment generated, and the yaw moment generated by the mass center shift, respectively. The detailed expressions are represented in Appendix A, Equations (A3) and (A4).

Asymmetric morphing is primarily used to generate the desired yaw angle through the rolling of the MAV, controlling its heading. However, it will also generate longitudinal coupling motion with the lateral motion, making the dynamic equation more complicated. Nevertheless, asymmetric

morphing will only cause gravity center change in the $y_b$-axis. The shift of gravity center $\Delta y_{cg}$ is expressed as

$$\Delta y_{cg} = \frac{\delta_a}{|\delta_a|} \cdot \frac{2m_a l \cos \delta_a}{m}. \tag{7}$$

Because $\delta_a \in [-30°, 30°]$, $\Delta y_{cg}$ does not exceed $\pm 1.8$ mm, which is almost negligible.

### 3.2.2. Dynamic Response Analysis of Asymmetric Morphing

Based on the established nonlinear lateral dynamics model Equation (6), the state variables changes of the MAV during asymmetric morphing were examined, and the additional inertia forces and moments effects caused by asymmetric morphing were analyzed to determine the impact on the motion of the MAV. Assuming that the MAV is in steady level flight with an initial state of $V = 20$ m/s and $\alpha = 4°$, dynamic response analyses were conducted for cases I to IV, which were the same as in Section 3.1.2. At the initial flight state, the left canard and wing begins to sweep $15°$ simultaneously at 0.1 s and then recover; later, the right canard and wing begins to sweep $15°$ simultaneously at 0.5 s and then recover. The changes in the motion parameters of the MAV under asymmetric morphing response are shown in Figure 6.

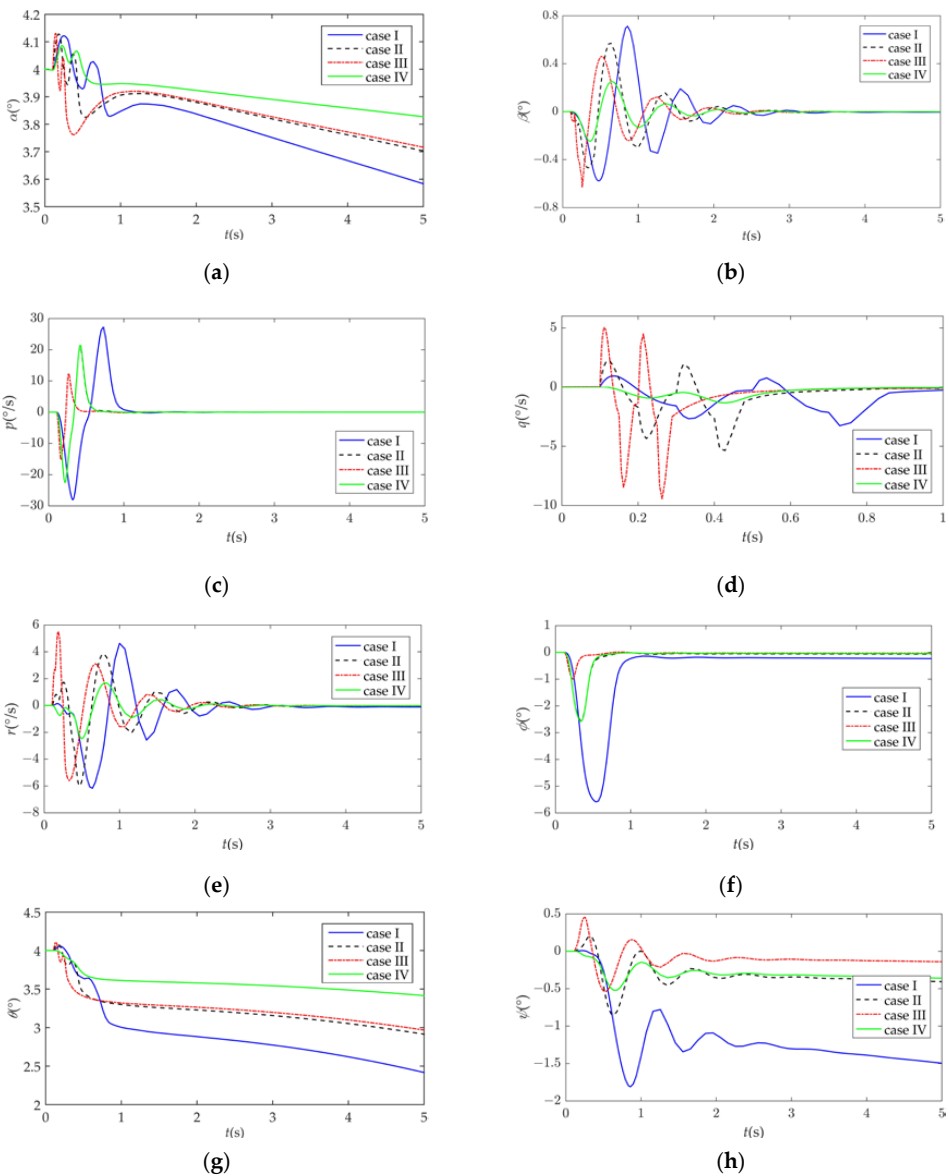

**Figure 6.** Dynamic responses of motion parameters caused by asymmetric morphing: (**a**) $\alpha$; (**b**) $\beta$; (**c**) $p$; (**d**) $q$; (**e**) $r$; (**f**) $\phi$; (**g**) $\theta$; (**h**) $\psi$.

It can be seen that when the left canard and wing of the MAV undergo sweeping ($\delta_a < 0$), the MAV rolls to left, and when the right canard and wing of the MAV undergo sweeping ($\delta_a > 0$), the MAV rolls to right. Roll rate $p$ and yaw angle $\psi$ also change accordingly. Since asymmetric morphing induces a pitch-down moment, the MAV undergoes minor pitch-down motion, and as time passes, both $\alpha$ and $\theta$ become less than their initial values. The rolling of the MAV also induces a left yawing motion. The sideslip angle $\beta$ and yaw rate $r$ oscillate, gradually approaching zero as the MAV returns to no roll angle, ultimately causing the heading of the MAV to deviate to the left. The influence of the morphing rate on the motion parameters of the MAV is similar to that of symmetric morphing. The faster the morphing rate, the quicker the response of the motion parameters of the MAV, and the smaller the amplitude of the overall change. The difference is that pitch rate $q$ and yaw rate $r$ show sudden changes in the early stage of asymmetric morphing, and the faster the morphing rate, the more obvious the changes in $q$ and $r$. Although this does not affect the overall trend, it causes more pronounced fluctuations in pitch and yaw movement. By comparing the response results of Case IV, it can be seen that this is mainly caused by the additional inertia forces and moments resulting from asymmetric morphing, which indicates that the additional force effect caused by asymmetric morphing mainly affects the yaw and pitch movements of the MAV and exacerbates the trend of pitch motion changes.

The change laws of the additional inertia forces and moments caused by asymmetric morphing with different morphing rates are shown in Figure 7.

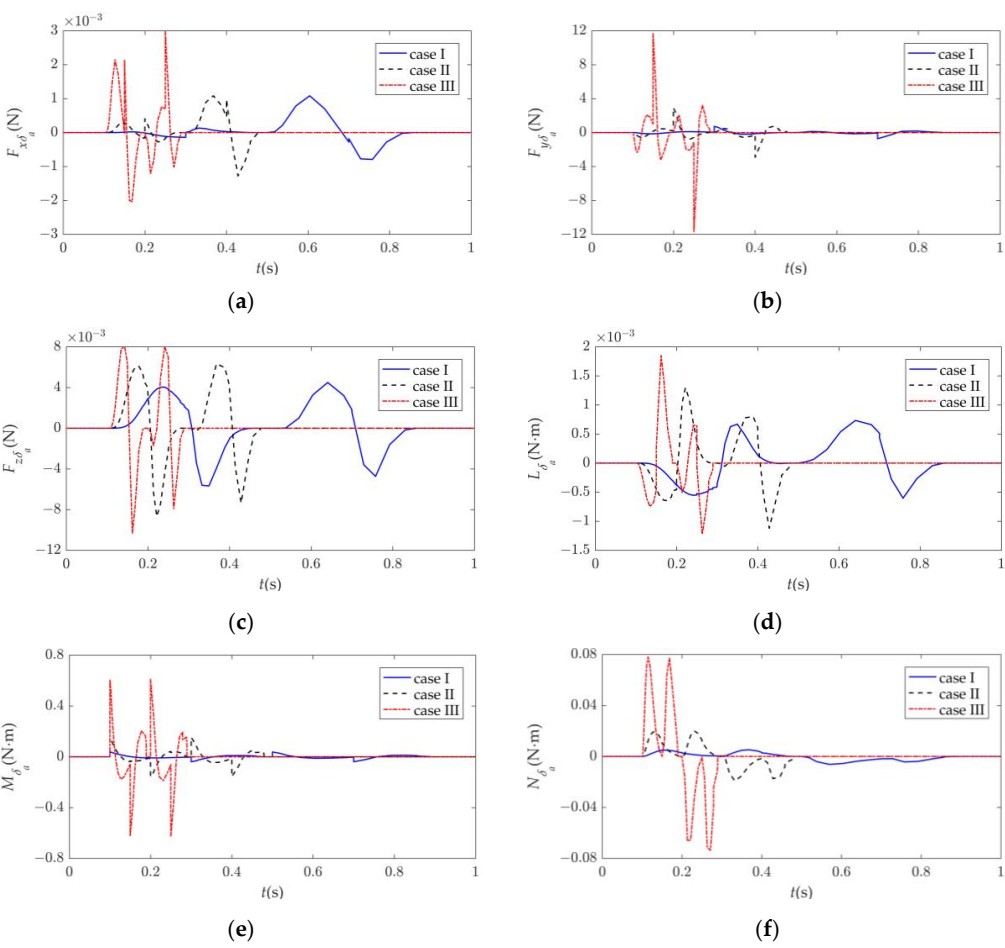

**Figure 7.** Additional inertia forces and moments caused by asymmetric morphing: (**a**) $F_{x\delta_a}$; (**b**) $F_{y\delta_a}$; (**c**) $F_{z\delta_a}$; (**d**) $L_{\delta_a}$; (**e**) $M_{\delta_a}$; (**f**) $N_{\delta_a}$.

It can be seen that the inertia forces and moments caused by asymmetric morphing also increase with the increase in morphing rate. The inertia forces are mainly generated in the $y_b$-axis, with a peak of 12 N, and the components in the $x_b$ and $z_b$ axes are approximately zero. The additional inertia roll moment is almost zero, and the additional pitch moment is more pronounced, with a maximum exceeding 0.6 N·m, while the additional yaw moment has a maximum of 0.08 N·m. This also explains

why the dynamic responses caused by asymmetric morphing show a more pronounced effect on yaw and pitch motions. Unlike the case of symmetric morphing, although the additional inertia force $F_{x\delta_a}$ is small, the inertia force $F_{y\delta_a}$ is significant. This is because when the canard and the wing sweep in opposite directions, the additional inertia forces in the $x_b$-axis generated by them cancel each other out. Additionally, because the canard is below and the wing is above the mass center of the MAV, the inertia forces generate a moment of coupling, resulting in a noticeable pitch moment. At the onset and end of the morphing, due to the sudden acceleration of the actuator, the changes in additional inertia forces and moments still have significant protrusions, but due to continuous changes in direction, their overall impact is not as significant as the peak values of the forces and moments suggest. Overall, the inertia forces and moments suppress the motion of the morphing control while exacerbating the changes in coupled movement. However, according to our analysis, adjusting the position of the canards and wings on the fuselage, namely, the distance $c$, can reduce the additional pitch moment generated by asymmetric morphing, thus reducing the coupling influence.

## 4. Experiments

An experimental system was set up to conduct dynamic response experiments and open-loop flight experiments. The experimental system of the MAV mainly consists of the tandem-wing MAV, elastic catapult, ground station equipment, and remote controller, as shown in Figure 8a. The MAV launches via a catapult in order to achieve faster takeoff speed and better takeoff stability. The electronic bay of the MAV is shown in Figure 8b, which includes the flight control system, the power module and propulsion system, servo actuators for airfoils, the data transport module, GPS, the video system, the air speedometer, and so on.

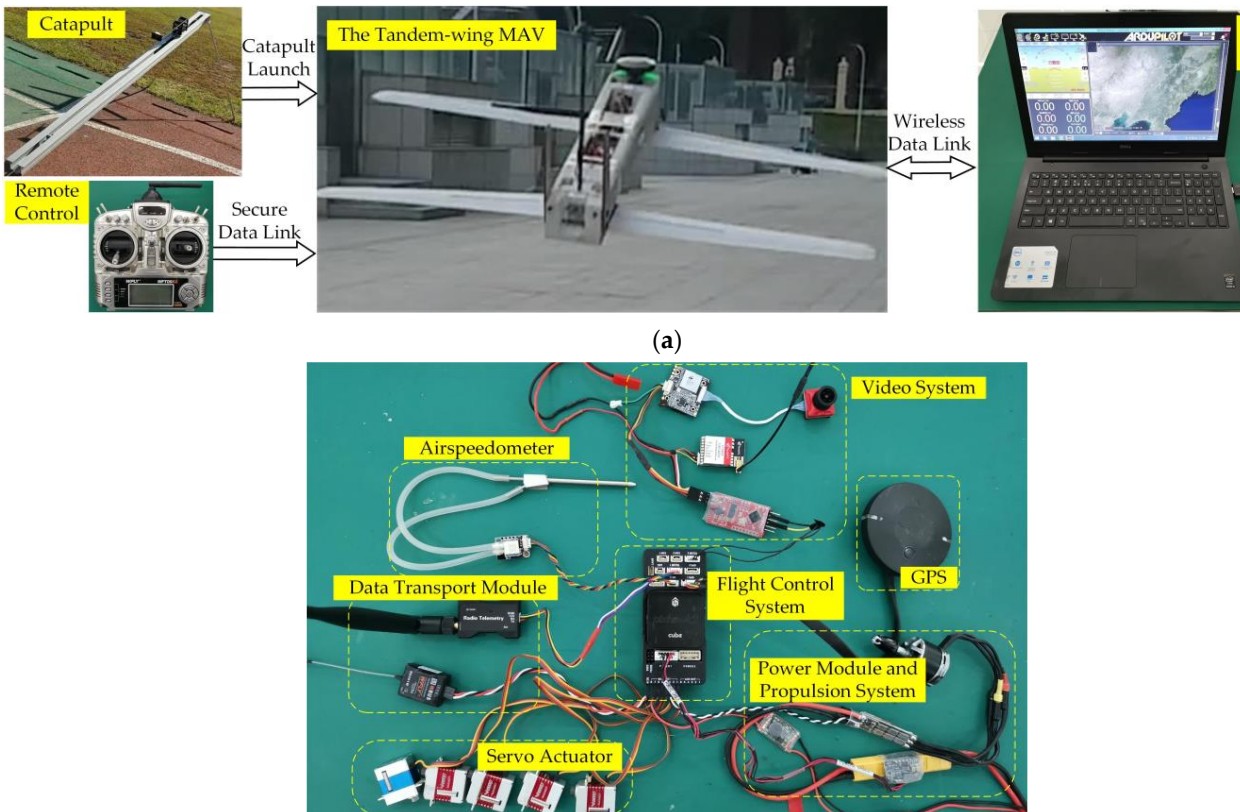

**Figure 8.** Experimental system construction: (**a**) the composition of the MAV experimental system; (**b**) electronics bay of the MAV.

### 4.1. Dynamic Response Experiments

In order to observe the effects of the additional inertia forces and moments caused by morphing on the MAV more intuitively, the MAV was suspended at the gravity center with a string and controlled to undergo symmetric and asymmetric morphing in open-loop conditions. The changes

in motion of the MAV were observed, attitude response data were collected, and the results were compared with those of a single wing undergoing sweep morphing.

Figure 9a shows the changes in motion of the MAV when $\delta_e$ changed from 0 to 30° and then back to 0°, with the time sequence of the images being 0 s, 0.29 s, 0.5 s, and 0.71 s. It can be seen that at the beginning of the morphing process, the MAV lowered its head due to the influence of additional inertia forces and the forward movement of the mass center. This indicated that the dynamic effects of symmetric morphing were opposite to the expected direction of motion during symmetric morphing control process. Combined with the response data (blue line), it could be seen that the change in $\theta$ was consistent with the morphing frequency and became more obvious as the morphing reached its maximum peak, with a maximum peak exceeding 6°. In addition, the changes in $\phi$ and $\psi$ had only tenuous connections with the morphing frequency. When the morphing direction was reversed, $\theta$ changed to the opposite direction. After morphing, the attitude of the MAV oscillated around the pitch axis while maintaining relatively stable horizontal movement, indicating that symmetric morphing had a significant influence on pitch motion.

Figure 9b shows the changes in motion of the MAV when $\delta_a$ changed from 0 to 30° and then back to 0°, with the time sequence of the images being 0 s, 0.25 s, 0.46 s, and 0.87 s. It can be seen that there was a continuous oscillatory motion around the roll axis when the MAV underwent asymmetric morphing. Due to the limitations of the experimental string, the amplitude of oscillation was not large. Combined with the response data (black line), it could be seen that the roll angle $\varphi$ and yaw angle $\psi$ changed in accordance with $\delta_a$. When $\delta_a$ reached its maximum peak, $\varphi$ and $\psi$ also produced peak fluctuations in the opposite direction to the expected direction of motion. With asymmetric morphing, the MAV had a minor yaw motion due to the difference in the distance from the canards and the wings to the center of mass. When the morphing direction changed, the yaw direction also changed. In addition, continuous asymmetric morphing did not increase the maximum change in each attitude angle of the MAV, and the range of change in $\theta$ was very small, with a value not exceeding $\pm 0.8°$. This was much smaller than the range of change when subjected to symmetric morphing. All of the above phenomena indicated that the additional inertia forces and moments produced by asymmetric morphing had a significant influence on the lateral motion of the MAV but only a minor effect on longitudinal motion.

Figure 9c shows the changes in motion of the MAV when the left canard underwent sweep morphing from 0 to 30° and then back to 0°, with the time sequence of the images being 0 s, 0.41 s, 0.83 s, and 1.08 s. Through the experiments and attitude data, it could be seen that when only a single wing was morphing, all three attitude angles of the MAV underwent significant changes. Due to the single-wing motion, the amplitude of changes in $\theta$ and $\varphi$ were smaller than those of symmetric and asymmetric morphing, respectively. However, the dynamic effects caused by a single wing morphing produced a more significant longitudinal and lateral coupled motion, and the change in $\psi$ was more evident. Obviously, the coupled motion resulting from single-wing sweep morphing is more difficult to control, while the dynamic effects caused by the sweep morphing of the tandem-wing MAV are only effective in a single direction of motion. This is why the tandem-wing MAV with multiple variable sweep wings can control the flight attitude and fly without an elevator and rudder.

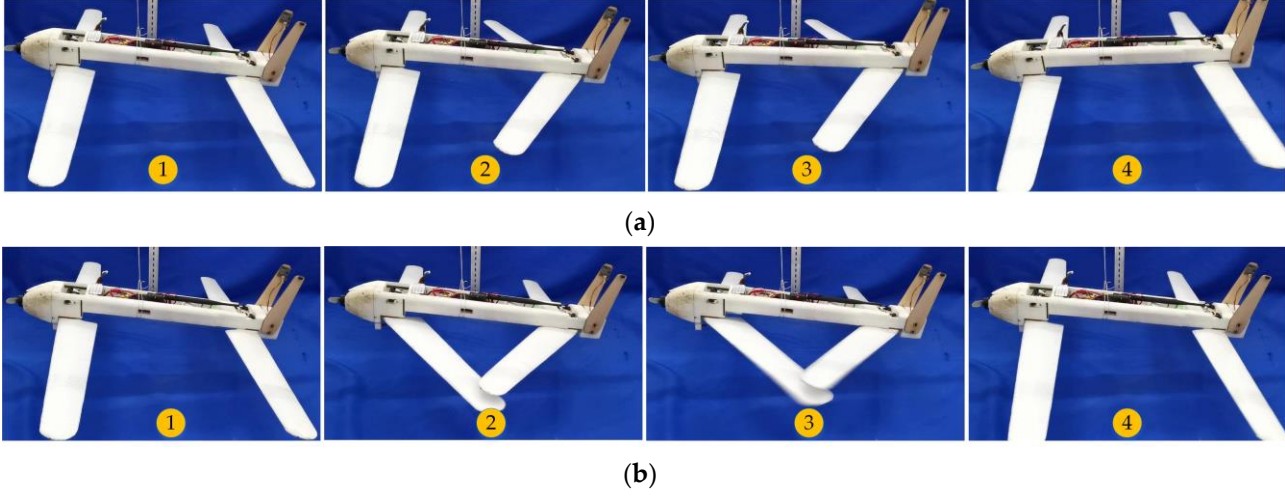

**Figure 9.** *Cont.*

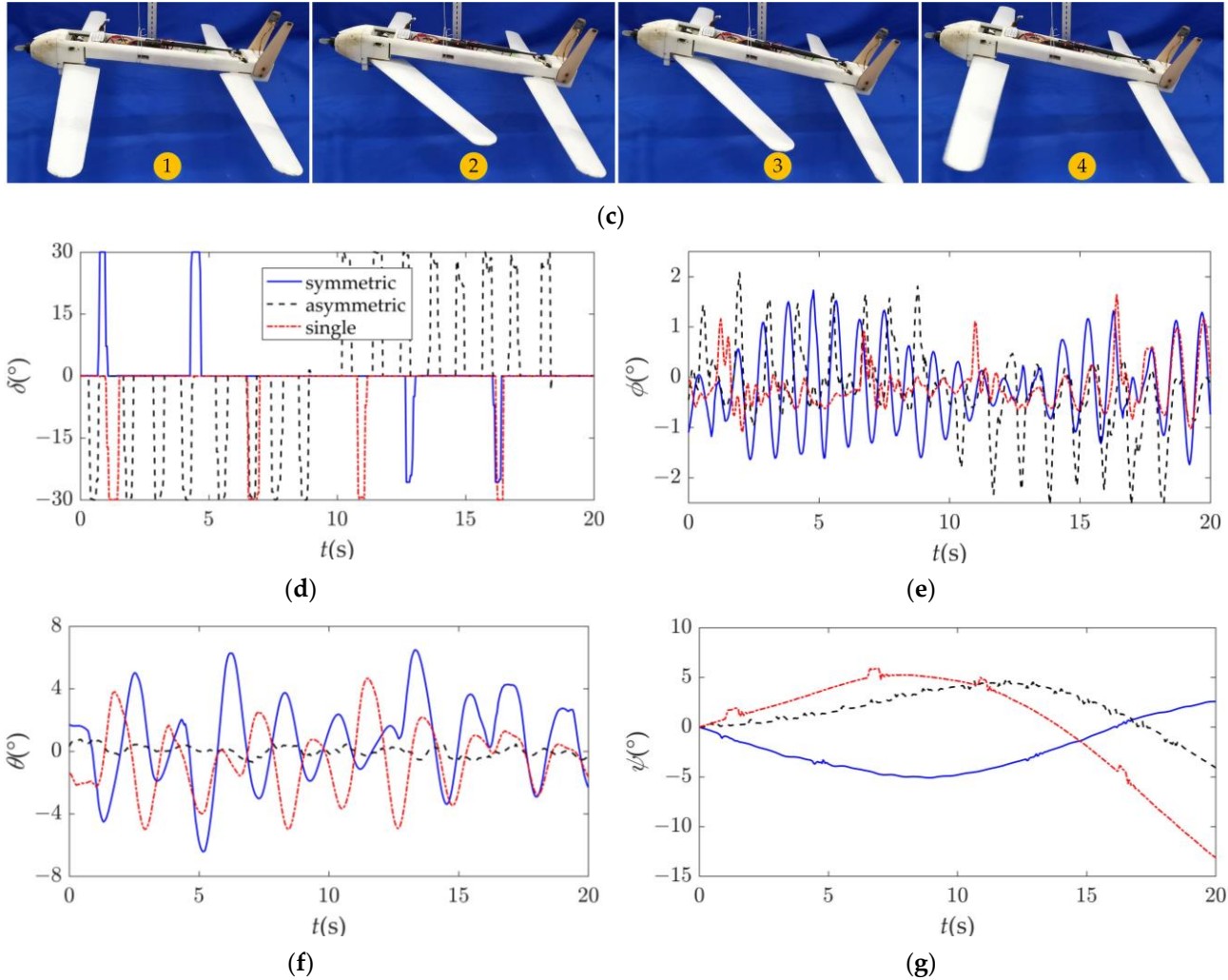

**Figure 9.** Dynamic response tests in different morphing conditions: (**a**) symmetric morphing; (**b**) asymmetric morphing; (**c**) a single wing sweep morphing; (**d**) $\delta$; (**e**) $\phi$; (**f**) $\theta$; (**g**) $\psi$.

### 4.2. Flight Experiments

The flight experiments in this study were conducted in an open field with a light wind using a remote controller in manual mode; the flight control system was mainly used to record the flight data and did not participate in flight control.

Figure 10 shows the flight process and flight data of the tandem-wing MAV using symmetrical morphing for climb control after catapult launching. From the flight images, it can be clearly seen that when the wings underwent symmetrical forward sweep morphing, the MAV transitioned from a level flight to a climbing state, with a quick control response. Combined with the flight data, it can be observed that when the MAV was climbing, the control input $\delta_e$ was greater than zero, and as the MAV climbed, the pitch angle $\theta$ was gradually controlled to stabilize. In addition, within 5 s after launching, the MAV quickly climbed to a flight altitude of about 10 m and achieved a flight speed of approximately 16 m/s. The experimental results demonstrate that symmetric morphing for pitch control of the tandem-wing MAV is feasible and has sufficient control effectiveness.

Figure 11 shows the flight process and flight data of the tandem-wing MAV using asymmetrical morphing for roll control. From the flight images, it can be seen that when the left canard and wing of the MAV underwent sweep morphing while the right canard and wing remains unchanged, the MAV quickly rolled to the left, while still maintaining a stable attitude flight. By analyzing the flight data, it was observed that the flight data during the roll control process almost matched with the simulation results. Under the continuous asymmetric morphing commands for left roll control, the MAV gradually yawed towards left, and at 10 s, the heading deviation was approximately 125°. During this process, the MAV also exhibited a slight nose-down tendency; ultimately, the flight

altitude decreased by about 2 m, which was fine-tuned through pitch control, while the flight speed remained at around 18 m/s. The experimental results demonstrate that asymmetric morphing for roll control of the tandem-wing MAV is feasible and has good control effectiveness.

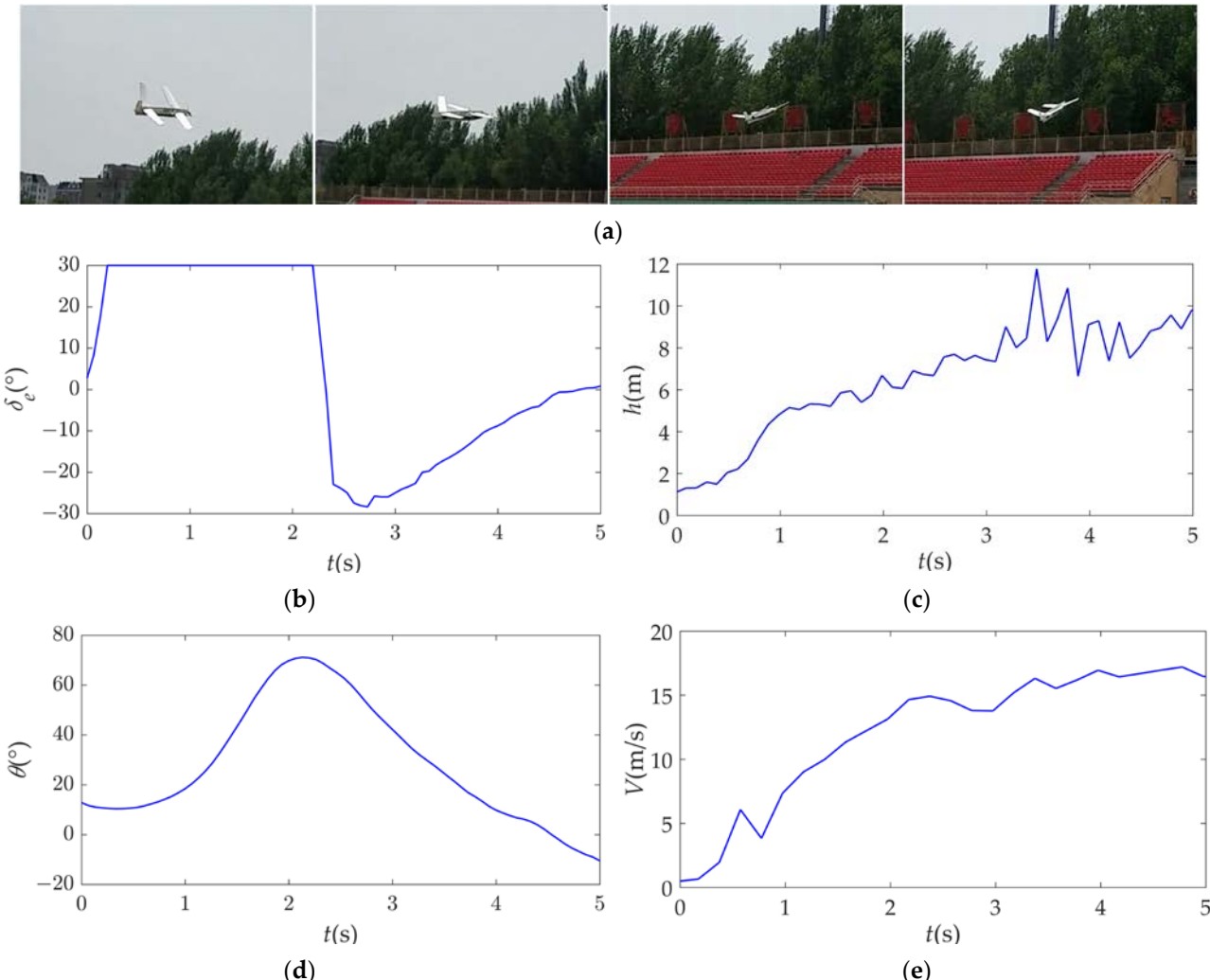

**Figure 10.** Flight test of symmetric morphing for pitch control: (**a**) pitch control test; (**b**) $\delta_e$; (**c**) $h$; (**d**) $\theta$; (**e**) $V$.

Through multiple flight experiments with the MAV, it was found that the roll and pitch control effects resulting from symmetrical morphing and asymmetrical morphing are similar to those obtained in simulations. However, the attitude data curve of the MAV had slight fluctuations. This was due to vibration interference caused by the rotating propellers and sweep morphing during the flight, leading to measurement errors in the attitude sensors. Additionally, the MAV was also affected by complex and varying external wind fields. In the future, it will be necessary to design control system with strong robustness in order to cope with external disturbances and improve flight quality, and further research on aerodynamic characteristics of the tandem-wing MAV during take-off stage is also needed in order to increase the success rate of catapult launching and enhance catapult stability. It should be noted that the morphing mode studied in this paper is not entirely superior to other morphing modes; it also has its limitations, such as structural complexity, additional weight, and increased number of servos. The use of this mode in this paper merely constituted an exploration into morphing modes, and we hope that this paper can stimulate divergent thinking in readers.

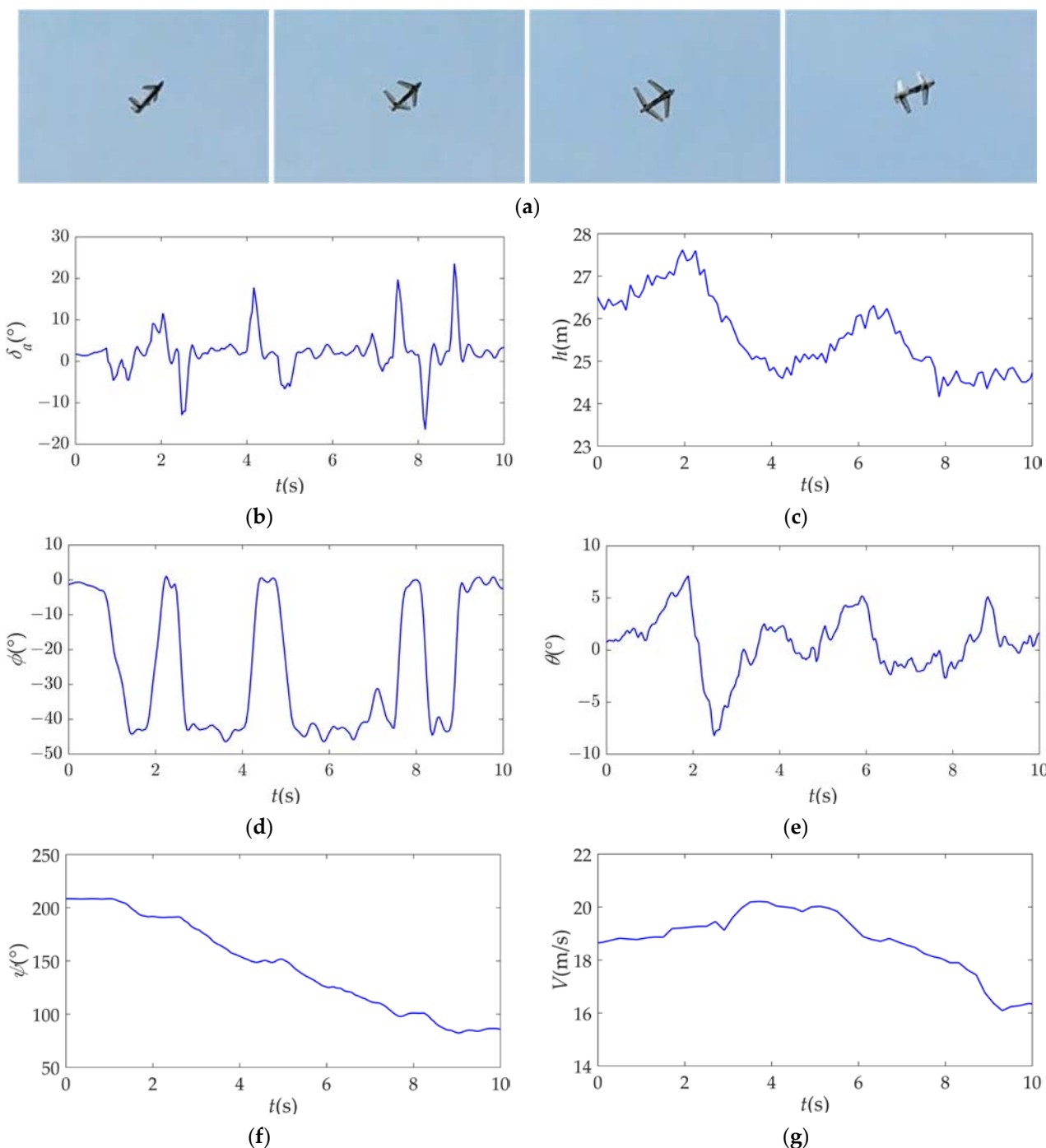

**Figure 11.** Flight test of asymmetric morphing for roll control: (**a**) roll control test; (**b**) $\delta_a$; (**c**) $h$; (**d**) $\phi$; (**e**) $\theta$; (**f**) $\psi$; (**g**) $V$.

## 5. Conclusions

In this paper, a tandem-wing MAV with multiple variable-sweep wings was proposed, the symmetric morphing of which was used for pitch control and the asymmetric morphing of which for roll control, realizing flight attitude control of the MAV without the traditional control surface. A prototype of the MAV was designed, and the load experiments verified that the structure had sufficient strength to support the wing flight loads. The change rules of the control moments caused by morphing were presented through numerical simulations, and the pitch and roll moments generated by morphing were considerable. Based on the previously established Kane dynamic equations, the longitudinal and lateral dynamic models were decoupled and simplified, and the effects on the MAV caused by symmetric and asymmetric morphing were investigated through dynamic response

simulation. The simulation results showed that when the MAV underwent symmetric morphing, the dynamic effects resulted in fluctuations in the state variables of the MAV, but also a certain inhibitory effect on the longitudinal motion, and the additional inertia forces were mainly in the $x_b$-axis. When the MAV underwent asymmetric morphing, the additional inertia forces were mainly in the $y_b$-axis, and the dynamic effect mainly affected the yaw and pitch motions of the MAV. The additional pitch inertia moment exacerbated the changes in coupled movement; however, adjusting the distance $c$ between the canards and wings could reduce the additional pitch moment, thus reducing the coupling influence. Then, dynamic response experiments were conducted, and the experimental results demonstrated that symmetric morphing had a significant influence on the pitch motion, and asymmetric morphing had a significant influence on the lateral motion but only a minor effect on the longitudinal motion, which verified the accuracy of the simulation results. Finally, the results of flight experiments demonstrated that the morphing mode in this study was feasible and effective for attitude control without the traditional aileron and elevator while flying.

**Author Contributions:** Conceptualization, L.G.; formal analysis, L.G.; funding acquisition, Y.Z.; investigation, Y.Z. and X.Z.; methodology, L.G. and Y.Z.; supervision, L.L. and J.Z. (Jie Zhao); validation, J.Z. (Junming Zhang) and B.C.; writing—original draft, L.G.; writing—review and editing, X.Z. All authors have read and agreed to the published version of the manuscript.

**Funding:** This research was funded by the National Outstanding Youth Science Fund Project of National Natural Science Foundation of China, grant number 52025054, and the China Postdoctoral Science Foundation, grant number 2021M690831.

**Data Availability Statement:** Not applicable.

**Conflicts of Interest:** The authors declare no conflict of interest.

## Abbreviations

| | |
|---|---|
| $\delta_e$ | command of symmetric morphing |
| $\delta_a$ | command of asymmetric morphing |
| $h_i$ | sweep angle of each airfoil |
| $\alpha$ | attack angle |
| $\beta$ | sideslip angle |
| $m$ | mass of the entire MAV |
| $m_a$ | mass of a single airfoil |
| $u$ | flight velocity in the $x_b$-axis |
| $v$ | flight velocity in the $y_b$-axis |
| $w$ | flight velocity in the $z_b$-axis |
| $\phi$ | roll angle |
| $\theta$ | pitch angle |
| $\psi$ | yaw angle |
| $p$ | roll angular velocity |
| $q$ | pitch angular velocity |
| $r$ | yaw angular velocity |
| $V$ | flight velocity |
| $L$ | lift |
| $Y$ | side force |
| $D$ | drag |
| $P$ | thrust |
| $L$ | roll moment |
| $M$ | pitch moment |
| $N$ | yaw moment |
| $F_x$ | forces in the $x_b$-axis |
| $F_y$ | forces in the $y_b$-axis |
| $F_z$ | forces in the $z_b$-axis |
| $\Delta x_{cg}$ | shift of gravity center in the $x_b$-axis |
| $\Delta y_{cg}$ | shift of gravity center in the $y_b$-axis |
| $H$ | height |

## Appendix A

The expressions of the additional inertia forces and moments caused by symmetric morphing are represented as follows:

$$
\begin{cases}
F_{x\delta_e} = 2m_a l(\dot{h}_1^2 \sin h_1 - \ddot{h}_1 \cos h_1 + \dot{h}_3^2 \sin h_3 - \ddot{h}_3 \cos h_3) \\
F_{z\delta_e} = 2m_a q l(\dot{h}_1 \cos h_1 + \dot{h}_3 \cos h_3) + 2m_a \dot{q} l(\sin h_1 + \sin h_3) \\
M_{\delta_e} = 2m_a q l(\dot{h}_3 l_2 \cos h_3 - \dot{h}_1 l_1 \cos h_1) + 2m_a l(\dot{w} - uq)(\sin h_1 + \sin h_3) \\
\quad - 2m_a c l(\ddot{h}_1 \cos h_1 - \ddot{h}_3 \cos h_3 - \dot{h}_1^2 \sin h_1 + \dot{h}_3^2 \sin h_3) \\
M_G = -2m_a g l \cos\theta(\sin h_1 + \sin h_3)
\end{cases}
\tag{A1}
$$

and

$$
\begin{cases}
l_1 = a_c + l \sin h_1 \\
l_2 = a_c - l \sin h_2 \\
l_3 = a_w - l \sin h_3 \\
l_4 = a_w + l \sin h_4 \\
J_y = J_y^f + 4J_y^a + 4m_a c^2 + m_a(l_1^2 + l_2^2 + l_3^2 + l_4^2)
\end{cases}
\tag{A2}
$$

where $a_c$ is the distance in the $x_b$-axis between the mass center of the fuselage and the canard rotation center; $a_w$ is the distance in the $x_b$-axis between the mass center of the fuselage and the wing rotation center; $c$ is the distance in the $z_b$-axis between the mass center of airfoil and of fuselage; $J_y^f$ is the moment of inertia of fuselage; and $J_y^a$ is the moment of inertia of airfoil. Detailed values are shown in Ref. [30], Table 1.

The expressions of the additional inertia forces and moments caused by asymmetric morphing are represented as follows:

$$
\begin{cases}
F_{x\delta_a} = 2m_a[-l\dot{r}\cos h_1 + lr\dot{h}_1 \sin h_1 + l\cos h_2 \dot{r} - lr\dot{h}_2 \sin h_2] \\
F_{y\delta_a} = -2m_a l(\ddot{h}_1 \sin h_1 - \ddot{h}_2 \sin h_2) - 2m_a l(\dot{h}_1^2 \cos h_1 - \dot{h}_2^2 \cos h_2) \\
F_{z\delta_a} = -2m_a l\dot{p}(-\cos h_1 + \cos h_2) - 2m_a p l(\dot{h}_1 \sin h_1 - \dot{h}_2 \sin h_2) \\
\overline{L}_{\delta_a} = 2m_a c l r(\dot{h}_1 \cos h_1 - \dot{h}_2 \cos h_2) + 2m_a l p[\dot{h}_1 \sin h_1(b_f + l \cos h_1) + \dot{h}_2 \sin h_2(b_f + l \cos h_2)] \\
\quad + 2m_a l(a_c - a_w)(-\cos h_1 + \cos h_2)\dot{q} - 2m_a l(\dot{w} + vp - uq)(\cos h_2 - \cos h_1) + 2m_a c l(\sin h_1 - \sin h_2)\dot{r} \\
\overline{L}_G = 2m_a g l \cos\theta \cos\varphi(\cos h_2 - \cos h_1) \\
M_{\delta_a} = -2m_a l c(\ddot{h}_1 \cos h_1 - \ddot{h}_2 \cos h_2 - \dot{h}_1^2 \sin h_1 + \dot{h}_2^2 \sin h_2) - m_a l(a_c - a_w)(\cos h_1 - \cos h_2)\dot{p} \\
\quad + m_a l p(a_c - a_w)(\dot{h}_1 \sin h_1 - \dot{h}_2 \sin h_2) - m_a l q(a_c + a_w)(\dot{h}_1 \cos h_1 - \dot{h}_2 \cos h_2) - 2m_a q l^2(\dot{h}_1 \sin h_1 + \dot{h}_2 \sin h_2) \\
N_{\delta_a} = -m_a l(\dot{h}_1^2 \cos h_1 - \dot{h}_2^2 \cos h_2)(a_c - a_w) - m_a l r(a_c + a_w)(\dot{h}_1 \cos h_1 - \dot{h}_2 \cos h_2) \\
\quad + 2m_a l b_f r(\dot{h}_1 \sin h_1 + \dot{h}_2 \sin h_2) + 2m_a c l(\sin h_1 - \sin h_2)\dot{p} - 2m_a l(\dot{u} + wq - vr)(\cos h_1 - \cos h_2) \\
N_G = 2m_a g l \sin\theta(\cos h_2 - \cos h_1)
\end{cases}
\tag{A3}
$$

and

$$
\begin{cases}
J_x = J_x^f + 4J_x^a + 4m_a c^2 + 2m_a[(b_f + l \cos h_1)^2 + (b_f + l \cos h_2)^2] \\
J_z = J_z^f + 4J_z^a + m_a(l_1^2 + l_2^2 + l_3^2 + l_4^2) + 2m_a[(b_f + l \cos h_1)^2 + (b_f + l \cos h_2)^2] \\
J_1 = J_z^f - J_y^f + 4(J_z^a - J_y^a), \\
J_2 = J_x^f - J_z^f + 4(J_x^a - J_z^a) \\
J_3 = J_y^f - J_x^f + 4(J_y^a - J_x^a)
\end{cases}
\tag{A4}
$$

where $b_f$ is the distance in the $y_b$-axis between the mass center of the fuselage and the airfoil rotation center; and $J_i^j$ is the moment of inertia of fuselage and airfoil. Detailed values are shown in Ref. [30], Table 1.

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
