# Peer review of "Dynamic Analysis and Experiment of Multiple Variable Sweep Wings on a Tandem-Wing MAV"

_drones, doi:10.3390/drones7090552_

Round 1

Reviewer 1 Report

The paper deal with a tandem-wing micro aerial vehicle (MAV) with multiple variable-sweep wings instead of traditional aileron and elevator. It is justified that such a tandem-wing MAV can morph symmetrically for pitch control and asymmetrically for roll control. For this reason, the proposed MAV design seems to be perspective.

The paper under consideration is the continuation of the previous paper by the authors [25], where the dynamic model of the tandem-wing MAV has been built. As a result, section 3 (Modeling and Dynamic Analysis) of the present paper seems to be weak since it cannot be understood without paper [25].

The review is relevant and complete. No gaps in knowledge are identified. The references are appropriate.

The remarks on the manuscript are as follows

1) Page 3, line 118: abbreviation EPP is not explained.

2) Page 5, Fig. 3 (b): the difference between 0 and -0 is not clear.

3) Page 6: x_b, y_b, z_b axes are neither explained, nor shown at the figures.

4) Pages 5, 6, ...: The terms "roll moment", "pitch moment", "gravity moment" are not clear until the pole for moment calculation is not specified. The choice of the pole is necessary in the definition of the moment of force.

5) Page 6, line 198: The expression "gravity moment generated by the mass center shift" is not clear until two points specified: the pole for moment calculation and the starting point for shift calculation. 

My opinion is that the paper can be published after revision.

Author Response

Dear Reviewer,

Thank you very much for your valuable feedback that we have used to improved our previous manuscript. We have addressed the comments and made modifications and supplements to our manuscript.

Please see the attachment for point by point responses.

Reviewer 2 Report

The very premise of the study is woefully deficient. The authors must review actual morphing technologies. They will see that the state-of-the-art was set nearly 15 years ago as flight aircraft with true morphing-wing UAVs took to the air and demonstrated high order benefits like dramatically increased maneuverability, lower power consumption, inherent gust rejection, improved maximum lift, reduced stall speeds, higher bandwidth and lower weight fraction devoted to the flight control system. The work of Vos represents the best performance adaptive wing technologies for UAVs and has flown many times showing high level benefits. Barrett authored several papers which sum up the state of the art of adaptive aerostrutures in flying aircraft, including and especially the first adaptive UAVs dating to 1994 and 1997 for fixed- and rotary-wing UAVs. That none of these important papers are referenced indicates a gross lack of understanding of the field as a whole. If the authors wish to have a chance to publish, these and other real adaptive aerostructures papers must be properly referenced.  The authors' approach is clearly heavier, has extremely poor response time, dangerously alters the static margin and is all set in a terrible configuration. There are reasons why tandem-wing aircraft are few and far between... and they are almost all related to flight safety and performance. It's a very bad configuration that  should never have been explored. The authors must tell the reader point blank that this very concept is catastrophically flawed. Truth be told, it would be best for the sake of the reputations of the authors simply to withdraw the paper. If the paper proceeds, it could be a professional embarrassment to the authors... 

The English is suitable, it's just that the concept is catastrophically flawed. 

Author Response

(The authors gave the same response as above.)

Reviewer 3 Report

The paper covers dynamic analysis of variable sweep tandem-wing MAV with details of dynamic modeling and experiments to support the the assertion that by sweeping the canards and the wings in opposite directions or asymmetrically, the additional inertia forces and moments that would otherwise be generated using morphing the wings without sweep, could be counteracted thus reducing the coupling between lateral and longitudinal dynamics and perhaps lead to simplified control system. It is an extension of their earlier work in which they investigated longitudinal dynamics (Ref. 24) and both longitudinal and lateral dynamics during catapult-launch (Ref. 25). This idea presents an attractive alternative to the extensive current focus on shape/profile morphing techniques that introduce significant inertial forces and moments presenting additional challenges for control system design. Although, the sweep morphing concept also results in additional inertia forces and moments, however, they interact to weaken the coupling between longitudinal and lateral motion during morphing. Therefore, it is setting a novel direction for focus that can simplify the design of control systems once the subject is fully understood. Hence, this paper will add value to the existing literature and therefore is highly recommended for publishing.

The paper presents both dynamic modeling (longitudinal and lateral) as well as experiments to determine the dynamic response to symmetric and asymmetric sweep morphing under different conditions (rates and amplitudes). It would be good to add some results to show the effect of relative position of wing and canard with respect to the mass center on the resulting pitcing and roll moments and perhaps there is a posiion where the additional pitch and roll are effectively cancelled thus reducing further the control system requirements.

The authors state that the flight data was close to the simulation results. It is suggested to show the comparison between the simulation and experiments based on the experimental run conditions or data to substantiate the accuracy of the model and also to strengthen the assertion that sweep morphing has significant advantages that can help simplify the control design design.

Lastly, some minor editing of English is recommended. Some of the sentences have been highlighted in this regards to improve the language and understanding

Lastly, some minor editing of English is recommended. Some of the sentences have been highlighted in this regards to improve the language and understanding

Author Response

(The authors gave the same response as above.)
